# LIMITATIONS FOR LEARNING FROM POINT CLOUDS

## ABSTRACT

In this paper we prove new universal approximation theorems for deep learning on point clouds that do not assume fixed cardinality. We do this by first generalizing the classical universal approximation theorem to general compact Hausdorff spaces and then applying this to the permutation-invariant architectures presented in *PointNet* (Qi et al) and *Deep Sets* (Zaheer et al). Moreover, though both architectures operate on the same domain, we show that the constant functions are the only functions they can mutually uniformly approximate. In particular, DeepSets architectures cannot uniformly approximate the diameter function but can uniformly approximate the center-of-mass function but it is the other way around for PointNet. Additionally, even when the point clouds are limited to at most $k$ points, PointNet cannot uniformly approximate center-of-mass and we obtain explicit error bounds and a method to produce geometrically derived adversarial examples.

## 1 INTRODUCTION

Recently, architectures proposed in *PointNet* (Qi et al., 2017) and *Deep Sets* (Zaheer et al., 2017) have allowed for the direct processing of point clouds within a deep learning framework. These methods produce outputs that are permutation-invariant with respect to the member points and work for point clouds of arbitrarily large cardinality. A common source of such data is LIDAR measurements from autonomous vehicles. Zaheer et al. (2017) also presents a permutation-equivariant architecture which we do not discuss here.

Each of of these works provide their own universal approximation theorem (UAT) to support the empirical success of their architectures. However, both results assume the cardinality of the point cloud is fixed to some size $n$. In this work we refine these results, remove the cardinality limitation, use weaker architecture assumptions, and arrive at three main results which can be summarized roughly as follows (assuming unrestricted finite cardinality for the input point clouds):

1) PointNet (DeepSets) architectures can uniformly approximate real-valued functions that are uniformly continuous with respect to the Hausdorff (Wasserstein) metric and nothing else (Theorem 3.4).

2) Only the constant functions can be uniformly approximated by both architectures. In particular, PointNet architectures can uniformly approximate the diameter function but DeepSets architectures cannot. Conversely, DeepSets architectures can uniformly approximate the center-of-mass function but PointNet architectures cannot (Theorem 4.1).

3) We prove explicit error lower bounds and produce adversarial examples to show that even when limited to point clouds of size $k$, PointNet cannot uniformly approximate center-of-mass (Theorem 4.2).

To do this we extend the many universal approximation results for feed-forward networks (Cybenko, 1989; Hornik et al., 1989; Leshno et al., 1993; Stinchcombe, 1999) to the abstract setting of general compact Hausdorff spaces. We then find appropriate compact metric spaces over which PointNet and DeepSets architectures can be easily analyzed and then finally we observe the resulting consequences in the original setting of interest, i.e. point clouds.

## 2 Preliminaries

### 2.1 PointNet and DeepSets Architectures

In practice, the implementations of the architectures presented in *PointNet* and *Deep Sets* can involve many additional tricks, but the essential ideas are quite simple. We do however make a small modification to the *Deep Sets* model. For $A \subseteq \mathbb{R}^n$ of cardinality $|A| < \infty$, we have Qi et al. (2017) and Zaheer et al. (2017) suggesting scalar-output neural networks of the form

$$F_{PN}(A) = \rho \left( \max_{\boldsymbol{a} \in A} \boldsymbol{\varphi}(\boldsymbol{a}) \right), \quad \text{and} \quad F_{DS}(A) = \rho \left( \boldsymbol{b} + \frac{1}{|A|} \sum_{\boldsymbol{a} \in A} \boldsymbol{\varphi}(\boldsymbol{a}) \right),$$

respectively. Here $\boldsymbol{\varphi} : \mathbb{R}^n \to \mathbb{R}^m$ creates features for each point in $A$, then a symmetric operation is applied, and then $\rho : \mathbb{R}^m \to \mathbb{R}$ combines these features into a scalar output (here $\max$ is the component-wise maximum). In practice, we need both $\rho$ and $\boldsymbol{\varphi}$ to be neural networks. Note that because we use a symmetric operation before $\rho$, the output will not depend on the ordering of points in the point cloud, and because the max and sum operations scale to arbitrary finite cardinalities the size of the point cloud is not an issue. The original model in *Deep Sets* did not have a bias term $\boldsymbol{b}$ and used a sum instead of the averaging we use here. This change will help us later in our theoretical analysis.

It will help to introduce some simplifying notation. Let $\mathcal{F}(\Omega)$ denote the set of all nonempty finite subsets of a set $\Omega$ (i.e. point clouds in $\Omega$), $\mathcal{F}^{\leq k}(\Omega)$ the set of nonepmty subsets of size $\leq k$, and $\mathcal{F}^k(\Omega)$ the set of $k$-point subsets. Now consider $\Omega \subseteq \mathbb{R}^N$ and define $\max_f, \mathrm{ave}_{f,b} : \mathcal{F}(\Omega) \to \mathbb{R}$ which are given by $\max_f(A) = \max_{\boldsymbol{a} \in A} f(\boldsymbol{a})$ and $\mathrm{ave}_{f,b}(A) = b + \frac{1}{|A|} \sum_{\boldsymbol{a} \in A} f(\boldsymbol{a})$ respectively. We make sense of this in the natural way if we use vector-valued $\boldsymbol{f}$ and $\boldsymbol{b}$ by operating component-wise. We call these operations max neurons and biased-averaging neurons respectively.

Once again letting $\rho$ and $\boldsymbol{\varphi}$ be neural networks, $F_{PN} = \rho \circ \max_{\boldsymbol{\varphi}}$ and $F_{DS} = \rho \circ \mathrm{ave}_{\boldsymbol{\varphi}, \boldsymbol{b}}$ will be the general form of what we call the PointNet and DeepSets architectures (resp.) in this paper.

Some natural questions are 1) is there a topology for $\mathcal{F}(\Omega)$ that makes these architectures continuous, 2) how expressive are these approaches, and 3) how deep is deep enough for function approximation?

### 2.2 Function Spaces and Uniform Approximation

From now on, we only consider $\mathbb{R}$-valued functions unless otherwise stated. Let $\mathcal{B}(A)$ be the set of bounded functions on a set $A$, let $\mathcal{C}(X)$ and $\mathcal{C}_b(X)$ be the set of continuous and bounded continuous functions on a topological space $X$ (respectively), and let $\mathcal{U}(M)$ and $\mathcal{U}_b(M)$ be the uniformly continuous and bounded uniformly continuous functions on a metric space $(M, d)$ (respectively). We equip all of these with the uniform norm i.e. $\|f\|_A = \sup_{a \in A} |f(a)|$ – we reserve $\|*\|$ for the Euclidean norm. This makes them all normed spaces, with $\mathcal{B}(A)$, $\mathcal{C}_b(X)$ and $\mathcal{U}_b(M)$ additionally being Banach spaces. Moreover, if $X$ is compact and $(M, d)$ has compact metric completion, then $\mathcal{C}(X) = \mathcal{C}_b(X)$ and $\mathcal{U}(M) = \mathcal{U}_b(M)$ and hence are also Banach spaces. For background see Rudin (2006).

If given an injective map $i : A \to X$, then we say that $\varphi : A \to \mathbb{R}$ (uniquely) continuously extends to $X$ if there is a (unique) $\widetilde{\varphi} \in \mathcal{C}(X)$ such that $\widetilde{\varphi} \circ i = \varphi$. We say a family of funcntions $\mathcal{N}$ on $A$ (uniquely) continuously extends to $X$ if every $\varphi \in \mathcal{N}$ (uniquely) continuously extends to $X$.

We will make use of the following lemma which is proved in the appendix.

**Lemma 2.1.** *Let $\mathcal{N} \subseteq \mathcal{B}(D)$ where $D$ is a dense subset of a compact metric space $(X, d)$. Suppose $\mathcal{N}$ has a continuous extension to $X$ denoted by $\mathcal{N}' \subseteq \mathcal{C}(X)$ which is dense. Then the uniform closure of $\mathcal{N}$ in $\mathcal{B}(D)$ is $r(\mathcal{C}(X)) = \mathcal{U}(D)$ where $r : \mathcal{C}(X) \to \mathcal{C}_b(D)$ is the domain restriction map.*

Letting $D = \mathcal{F}(\Omega)$, this lemma suggest the following plan of attack: find a compact metric space $(X, d)$ in which we can realize $\mathcal{F}(\Omega)$ as a dense subset and hope that our class of neural networks $\mathcal{N}$ continuously extends to a dense subset of $\mathcal{C}(X)$. If we can do that, then we know the uniform closure of our class of neural networks are precisely the uniformly continuous functions on $\mathcal{F}(\Omega)$ with respect to the metric inherited from $X$. This motivates the next subsection.

## 2.3 Metrics on the Space of Point Clouds

From now on we will assume $(\Omega, d)$ is a compact metric space and when $\Omega \subseteq \mathbb{R}^n$ it will be compact and equipped with the Euclidean metric. Let $\mathcal{K}(\Omega)$ denote the set of all compact subsets of $\Omega$ and $\mathcal{P}(\Omega)$ denote the set of all Borel probability measures on $\Omega$. The Hausdorff metric $d_H$ (Munkres, 2000) is a natural metric for $\mathcal{K}(\Omega)$ and 1-Wasserstein metric $d_W$ (Villani, 2009) (also called the Earth-mover distance) is a natural metric for $\mathcal{P}(\Omega)$. With these metrics, $\mathcal{K}(\Omega)$ and $\mathcal{P}(\Omega)$ become compact metric spaces of their own. From now on we will assume these two spaces are always equipped with the aforementioned metrics.

We also briefly mention $M(\Omega)$ the Banach space of finite signed regular Borel measures on $\Omega$. By the Riesz-Markov theorem it is the topological dual space of $\mathcal{C}(\Omega)$. Of interest to us is that $\mathcal{P}(\Omega) \subseteq M(\Omega)$ and that the weak-* topology on $\mathcal{P}(\Omega)$ coincides with the topology induced by $d_W$. This means that $d_W(\mu_n, \mu) \to 0$ iff $\int f \, d\mu_n \to \int f \, d\mu$ for all $f \in \mathcal{C}(\Omega)$.

Next, note that $\mathcal{F}(\Omega) \subseteq \mathcal{K}(\Omega)$ and let $i_\mathcal{K}$ denote the natural inclusion map. We can also define an injective map $i_\mathcal{P} : \mathcal{F}(\Omega) \to \mathcal{P}(\Omega)$ by mapping $A \in \mathcal{F}(\Omega)$ to its associated empirical measure $i_\mathcal{P}(A) = \mu_A = \frac{1}{|A|} \sum_{a \in A} \delta_a \in \mathcal{P}(\Omega)$ where $\delta_a$ is the Dirac delta measure supported at $a$. The injective maps $i_\mathcal{K}$ and $i_\mathcal{P}$ allow us to induce the $d_H$ and $d_W$ metrics on $\mathcal{F}(\Omega)$. We will denote the metrized versions by $\mathcal{F}_H(\Omega)$ and $\mathcal{F}_W(\Omega)$ respectively and use the same convention for $\mathcal{F}_H^k(\Omega)$ and $\mathcal{F}_W^k(\Omega)$. Another important fact to know is that $i_\mathcal{K}$ and $i_\mathcal{P}$ embed $\mathcal{F}(\Omega)$ as dense subset of $\mathcal{K}(\Omega)$ and $\mathcal{P}(\Omega)$. The former follows from compactness of the members of $\mathcal{K}(\Omega)$ and to see why the latter is true see Fournier & Guillin (2015); Villani (2009)

For $f \in \mathcal{C}(\Omega)$ and $b \in \mathbb{R}$, define $\mathrm{Max}_f : \mathcal{K}(\Omega) \to \mathbb{R}$ and $\mathrm{Ave}_{f,b} : \mathcal{P}(\Omega) \to \mathbb{R}$ as the functions $\mathrm{Max}_f(K) = \max_{x \in K} f(x)$ and $\mathrm{Ave}_{f,b}(\mu) = b + \int_\Omega f \, d\mu$.

**Lemma 2.2.** *Let $(\Omega, d)$ be compact, $f \in \mathcal{C}(\Omega)$, and $b \in \mathbb{R}$. Then $\mathrm{Max}_f \in \mathcal{C}(\mathcal{K}(\Omega))$ and $\mathrm{Ave}_{f,b} \in \mathcal{C}(\mathcal{P}(\Omega))$ and $\mathrm{Max}_f \circ i_\mathcal{K} = \max_f$ and $\mathrm{Ave}_{f,b} \circ i_\mathcal{P} = \mathrm{ave}_{f,b}$. As a consequence, PointNet and DeepSets are uniformly continuous on $\mathcal{F}_H(\Omega)$ and $\mathcal{F}_W(\Omega)$ respectively.*

This lemma (proved in the appendix) tells us that the max neurons and biased-averaging neurons continuously extend to $\mathcal{K}(\Omega)$ and $\mathcal{P}(\Omega)$ and hence so do PointNet and DeepSets architectures (since we merely compose with the continuous $\rho$ after). Thus, we wil be able analyze such architectures as continuous functions on compact metric spaces, which is mathematically a much nicer problem than studying them as set-theoretic functions on an un-metrized $\mathcal{F}(\Omega)$.

## 2.4 Generalized Neural Network Notation

For $\mathcal{A}$ a collection of functions from $X$ to $Y$ and $\mathcal{B}$ a collection of functions from $Y$ to $Z$, we denote the set of all compositions by $\mathcal{B} \circ \mathcal{A} = \{f \circ g \mid f \in \mathcal{B}, g \in \mathcal{A}\}$. In the case of a single function $\sigma : Y \to Z$ we let $\sigma \circ \mathcal{A} = \{\sigma \circ f \mid f \in \mathcal{A}\}$ and similarly for right-composition.

Next, let Aff denote the set of all affine functionals on $\mathbb{R}^N$, i.e. any function of the form $f(\boldsymbol{x}) = \boldsymbol{w} \cdot \boldsymbol{x} + b$. Let $\mathcal{N}^\sigma := \mathrm{span}\{\sigma \circ \mathrm{Aff}\}$ denote the set of single hidden-layer neural networks with linear output-layer and activation function $\sigma$, and then denote an $H$-layered network by $\mathcal{N}^{\boldsymbol{\sigma}}$ where $\boldsymbol{\sigma} = (\sigma_1, \ldots, \sigma_H)$ is a list of $H$-many activation functions where $\mathcal{N}^{(\boldsymbol{\sigma}, \tau)} := \mathcal{N}^{\boldsymbol{\sigma}, \tau} := \mathrm{span}(\tau \circ \mathcal{N}^{\boldsymbol{\sigma}})$.

Next we define various classes of PointNet networks whose weight functions are themselves neural networks. Let $\mathcal{N}_{PN}^\sigma := \mathcal{N}_{PN}^{\sigma; \varnothing} := \mathrm{span}\{\max_f \mid f \in \mathcal{N}^\sigma\}$ then define $\mathcal{N}_{PN}^{\sigma; \tau} := \mathrm{span}\{\tau \circ \mathcal{N}_{PN}^\sigma\}$. Like before, we can inductively define deeper networks, but we can also use deeper weight networks as well to create $\mathcal{N}_{PN}^{\boldsymbol{\sigma}; \boldsymbol{\tau}}$ – thus we have two distinct notions of depth.

Next, we do the same for DeepSets. Let $\mathcal{N}_{DS}^\sigma := \mathcal{N}_{DS}^{\sigma; \varnothing} := \mathrm{span}\{\mathrm{ave}_{f,b} \mid f \in \mathcal{N}^\sigma, b \in \mathbb{R}\}$. Note that $\mathrm{ave}_{f,b} + \mathrm{ave}_{g,c} = \mathrm{ave}_{f+g, b+c}$ and $\alpha \, \mathrm{ave}_{f,b} = \mathrm{ave}_{\alpha f, \alpha b}$. Thus since $\mathcal{N}^\sigma$ is a linear space, taking the span has no effect and $\mathcal{N}_{DS}^\sigma = \{\mathrm{ave}_{f,b} \mid f \in \mathcal{N}^\sigma, b \in \mathbb{R}\}$. Going one layer deeper yields $\mathcal{N}_{DS}^{\sigma; \tau} := \mathrm{span}\{\tau \circ \mathcal{N}_{DS}^\sigma\}$ which gets us new functions. Like with PointNet, we can inductively develop deeper families in two ways.

By Lemma 2.2 we can extend all the operations of our neural networks to $\mathcal{K}(\Omega)$ and $\mathcal{P}(\Omega)$ in a natural way. This let's us talk about about PointNet networks on $\mathcal{K}(\Omega)$ and DeepSets networks on

$\mathcal{P}(\Omega)$ which we'll define analogously by replacing $\max_f$ with $\mathrm{Max}_f$ and $\mathrm{ave}_{f,b}$ with $\mathrm{Ave}_{f,b}$. Thus

$$\mathcal{M}_{PN}^{\sigma} = \mathrm{span}\left\{\mathrm{Max}_f \mid f \in \mathcal{N}^{\sigma}\right\}, \qquad\qquad \mathcal{M}_{PN}^{\sigma;\tau} = \mathrm{span}\left\{\tau \circ \mathcal{N}_{PN}^{\sigma}\right\},$$
$$\mathcal{M}_{DS}^{\sigma} = \mathrm{span}\left\{\mathrm{Ave}_{f,b} \mid f \in \mathcal{N}^{\sigma}, b \in \mathbb{R}\right\}, \qquad \mathcal{M}_{DS}^{\sigma;\tau} = \mathrm{span}\left\{\tau \circ \mathcal{M}_{DS}^{\sigma}\right\}.$$

As before, the linear structure of $\mathcal{N}^{\sigma}$ makes $\mathcal{M}_{DS}^{\sigma} = \{\mathrm{Ave}_{f,b} \mid f \in \mathcal{N}^{\sigma}\}$

## 3   MAIN RESULTS

### 3.1   TOPOLOGICAL UAT

Leshno et al. (1993) prove that $\mathcal{N}^{\sigma}$ with $\sigma \in \mathcal{C}(\mathbb{R})$ has universal approximation property iff $\sigma$ is not a polynomial. For this reason, we will say a $\sigma \in \mathcal{C}(\mathbb{R})$ is 'universal' if it is non-polynomial and denote the set of all such such functions by $\mathfrak{U}(\mathbb{R})$. Using this theorem and Stone-Weierstrass we prove a UAT for certain kinds of two-hidden-layer 'neural networks' on an abstract compact Hausdorff space.

Recall that a family of functions $S$ on $\Omega$ separates points if for any $x \neq y$ there is an $f \in S$ so that $f(x) \neq f(y)$.

**Theorem 3.1** (Topological-UAT). *Let $X$ be a compact Hausdorff space and $\sigma \in \mathfrak{U}(\mathbb{R})$. If $S \subseteq \mathcal{C}(X)$ separates points and contains a nonzero constant, then $\mathrm{span}(\sigma \circ \mathrm{span}\,S)$ is dense in $\mathcal{C}(X)$. Additionally, if $S$ also happens to be a linear subspace, then $\mathrm{span}(\sigma \circ S)$ is dense in $\mathcal{C}(X)$.*

*Proof.* Let $S$ and $\sigma$ satisfy the above and let $V = \mathrm{span}\,S$. Let $\mathrm{Alg}(V)$ denote the algebra generated by $V$, i.e. all possible finite products, sums and scalar multiples of the elements of $V$. Then $\mathrm{Alg}(V)$ is unital subalgebra of $\mathcal{C}(X)$ that seperates points. By the Stone-Weierstrass theorem $\mathrm{Alg}(V)$ is dense in $\mathcal{C}(X)$. Now let $F \in \mathcal{C}(X)$ and $\epsilon > 0$ be arbitrary. By density there is a $G \in \mathrm{Alg}(V)$ such that $|F(a) - G(a)| < \epsilon/2$ for all $a \in X$. Since $G \in \mathrm{Alg}(V)$ there is an $N$-variable polynomial $p$ and $\boldsymbol{s} = (s_1, \ldots, s_N)$ where $s_i \in S$, so that $G = p \circ \boldsymbol{s}$. Since all $s_i \in \mathcal{C}(X)$ and $X$ is compact, the image $\boldsymbol{s}(X) \subseteq \mathbb{R}^N$ is compact. By the classical UAT (Leshno et al., 1993), there exists an $\eta \in \mathcal{N}^{\sigma}$ such that $|p(\boldsymbol{x}) - \eta(\boldsymbol{x})| < \epsilon/2$ for all $\boldsymbol{x} \in \mathbb{R}^N$. Thus,

$$|F(a) - (\eta \circ \boldsymbol{s})(a)| \leq |F(a) - p(\boldsymbol{s}(a))| + |p(\boldsymbol{s}(a)) - \eta(\boldsymbol{s}(a))| < \epsilon/2 + \epsilon/2 = \epsilon$$

for every $a \in X$. Finally note that $\eta(\boldsymbol{s}(a)) = \sum_{i=1}^{m} a_i \sigma(\boldsymbol{w}_i \cdot \boldsymbol{s}(a) + b_i)$ for some $a_i, b_i \in \mathbb{R}$ and $\boldsymbol{w}_i \in \mathbb{R}^N$. Since $S$ constains a nonzero constant, $\mathrm{span}\,S$ contains every constant and so $\boldsymbol{w}_i \cdot \boldsymbol{s} + b_i \in \mathrm{span}\,S$. Thus $\eta \circ \boldsymbol{s} \in \mathrm{span}(\sigma \circ \mathrm{span}\,S)$ as desired.

Lastly, if $S$ is also linear subspace, then $S = \mathrm{span}\,S$ and so $\mathrm{span}(\sigma \circ S)$ is dense in $\mathcal{C}(X)$. $\qquad\square$

### 3.2   POINT CLOUD UAT

We have met almost all the conditions required to use the topological-UAT on $\mathcal{K}(\Omega)$ and $\mathcal{P}(\Omega)$. We just need to show that $\mathrm{Max}_f$ and $\mathrm{Ave}_{f,b}$ yield nonzero constants and can separate points even when we limit ourselves to $f \in \mathcal{N}^{\sigma}$.

**Lemma 3.2** (Separation Lemma). *Let $\Omega \subseteq \mathbb{R}^N$ be compact and $\sigma \in \mathfrak{U}(\mathbb{R})$. Then $S_{PN} = \{\mathrm{Max}_f \mid f \in \mathcal{N}^{\sigma}\}$ and $S_{DS} = \{\mathrm{Ave}_{f,b} \mid f \in \mathcal{N}^{\sigma}, b \in \mathbb{R}\}$ separate points and contain constants.*

*Proof.* Let $d$ denote the Euclidean distance. First note that the constant function $h = \sigma(c) \in \mathcal{N}^{\sigma}$ for some $c \in \mathbb{R}$. Since $\sigma$ is not a polynomial, there is a choice of $c$ for which $\sigma(c) \neq 0$. This means $\mathrm{Max}_h \in S_{PN}$ and $\mathrm{Ave}_{h,0} \in S_{DS}$ are both constant. Now we just need to show that $S_{PN}$ and $S_{DS}$ separate points.

($S_{PN}$ separates points): Let $A, B \in \mathcal{K}(\Omega)$ with $A \neq B$. Without loss of generality, $A \setminus B \neq \varnothing$ so choose $a \in A \setminus B$. Let $f(x) = \min\{1, d(x, B)/d(a, B)\}$ and note that $f(a) = 1$, $f(B) = \{0\}$ and $f(\Omega) = [0, 1]$. By the classical UAT (Leshno et al., 1993) $\mathcal{N}^{\sigma}$ is dense in $\mathcal{C}(\Omega)$, so there is a $g \in \mathcal{N}^{\sigma}$ so that $|f(x) - g(x)| < 1/2$ for all $x \in \Omega$. Note $\mathrm{Max}_g \in S_{PN}$ and that $\mathrm{Max}_g(A) > 1/2$ and $\mathrm{Max}_g(B) < 1/2$. Since $A$ and $B$ were arbitrary, this shows $S_{PN}$ separates point in $\mathcal{K}(X)$.

($S_{DS}$ separates points): Given $\mu_1, \mu_2 \in \mathcal{P}(\Omega)$ with $\mu_1 \neq \mu_2$, by the Hahn-Banach separation theorem there exists a weak-* continuous linear functional $L : M(\Omega) \to \mathbb{R}$ that separates them. Let $\delta = |L(\mu_1) - L(\mu_2)|$. The topological dual of $M(\Omega)$ with the weak-* topology is equivalent to $\mathcal{C}(\Omega)$ and so there is an $f \in \mathcal{C}(\Omega)$ so that $L(\eta) = \int f \, d\eta$ for all $\eta \in M(\Omega)$. Since $\mathcal{N}^\sigma$ is dense in $\mathcal{C}(\Omega)$ there is a $g \in \mathcal{N}^\sigma$ so the that $|f(x) - g(x)| < \delta/2$ for all $x \in \Omega$. Define $J(\eta) = \int g \, d\eta$. Then for all $\eta \in \mathcal{P}(\Omega)$ we have $|L(\eta) - J(\eta)| \leq \int |f - g| \, d\eta < \frac{\delta}{2} \int d\eta = \delta/2$. Applying the triangle inequality we obtain,

$$\delta = |L(\mu_1) - L(\mu_2)| \leq \underbrace{|L(\mu_1) - J(\mu_1)|}_{<\delta/2} + |J(\mu_1) - J(\mu_2)| + \underbrace{|J(\mu_2) - L(\mu_2)|}_{<\delta/2}$$

Thus $0 < |J(\mu_1) - J(\mu_2)|$ and so $J = \text{Ave}_{g,0} \in S_{DS}$ separates $\mu_1$ and $\mu_2$. Since $\mu_1$ and $\mu_2$ were arbitrary, it follows that $S_{DS}$ seperates points in $\mathcal{P}(\Omega)$. $\qquad\square$

The following theorems show that one hidden layer in the weight networks and one hidden layer of the of the other kind suffice to prove the universal approximation theorems for PointNet and DeepSets.

**Theorem 3.3.** *Let $\Omega \subseteq \mathbb{R}^N$ be compact and $\sigma, \tau \in \mathfrak{U}(\mathbb{R})$. Then $\mathcal{M}_{PN}^{\sigma;\tau}$ and $\mathcal{M}_{DS}^{\sigma;\tau}$ are dense in $\mathcal{C}(A)$ and $\mathcal{C}(B)$ respectively, where $A \subseteq \mathcal{K}(\Omega)$ and $B \subseteq \mathcal{P}(\Omega)$ are closed subsets.*

*Proof.* Recall $\mathcal{M}_{PN}^{\sigma;\tau} = \text{span}\{\tau \circ \text{span}\, S_{PN}\}$ and $\mathcal{M}_{DS}^{\sigma;\tau} = \text{span}\{\tau \circ S_{DS}\}$. Since $\mathcal{K}(\Omega)$ and $\mathcal{P}(\Omega)$ are compact metric spaces, $A$ and $B$ are compact Hausdorff. By Lemma 3.2 we know $S_{PN}$ and $S_{DS}$ separate points and contain nonzero constants and so the topological-UAT (Theorem 3.1) yields the desired result. $\qquad\square$

**Theorem 3.4** (Point-Cloud-UAT). *Let $\Omega \subseteq \mathbb{R}^N$ be compact. If $\sigma, \tau \in \mathfrak{U}(\mathbb{R})$, then the uniform closure of $\mathcal{N}_{PN}^{\sigma;\tau}$ and $\mathcal{N}_{DS}^{\sigma;\tau}$ within $\mathcal{B}(\mathcal{F}(\Omega))$ is $\mathcal{U}(\mathcal{F}_H(\Omega))$ and $\mathcal{U}(\mathcal{F}_W(\Omega))$ respectively.*

*Proof.* $\mathcal{F}_H(\Omega)$ and $\mathcal{F}_W(\Omega)$ are isometrically isomorphic to $i_{\mathcal{K}}(\mathcal{F}(\Omega))$ and $i_{\mathcal{P}}(\mathcal{F}(\Omega))$ which are in turn dense in $(\mathcal{K}(\Omega), d_H)$ and $(\mathcal{P}(\Omega), d_W)$. By Lemma 2.2 we have that $\mathcal{N}_{PN}^{\sigma;\tau}$ and $\mathcal{N}_{DS}^{\sigma;\tau}$ continuously extend to $\mathcal{K}(\Omega)$ and $\mathcal{P}(\Omega)$ as $\mathcal{M}_{PN}^{\sigma;\tau}$ and $\mathcal{M}_{DS}^{\sigma;\tau}$. By Theorem 3.3 we know $\mathcal{M}_{PN}^{\sigma;\tau}$ and $\mathcal{M}_{DS}^{\sigma;\tau}$ are dense in $\mathcal{C}(\mathcal{K}(\Omega))$ and $\mathcal{C}(\mathcal{P}(\Omega))$. Finally, by Lemma 2.1 we have the desired result. $\qquad\square$

It's worth noting that we could have used Stinchcombe's generalization of the UAT to the case of neural networks on compact subsets of locally convex spaces (Stinchcombe, 1999) to prove that $\mathcal{M}_{DS}^{\sigma;\tau}$ is dense in $\mathcal{C}(\mathcal{P}(\Omega))$ but we chose the above route for consistency of technique and to be self-contained.

We now prove as a corollary a refinement of the universal approximation theorems of Qi et al. (2017) and Zaheer et al. (2017), both of which applied to the the case of $k$-point point clouds. In this version of the theorem we are able to restrict the depth of the neural network to just two hidden layers. The proof is essentially the same as Theorem 3.4.

**Corollary 3.5.** *Let $\Omega \subseteq \mathbb{R}^N$ be compact. If $\sigma, \tau \in \mathfrak{U}(\mathbb{R})$, then the uniform closure of $\mathcal{N}_{PN}^{\sigma;\tau}$ and $\mathcal{N}_{DS}^{\sigma;\tau}$ within $\mathcal{B}(\mathcal{F}^k(\Omega))$ are $\mathcal{U}(\mathcal{F}^k(\Omega)_H)$ and $\mathcal{U}(\mathcal{F}^k(\Omega)_W)$ respectively.*

*Proof.* $\mathcal{F}_H^k(\Omega)$ and $\mathcal{F}_W^k(\Omega)$ are isometrically isomorphic to $i_{\mathcal{K}}(\mathcal{F}^k(\Omega))$ and $i_{\mathcal{P}}(\mathcal{F}^k(\Omega))$ which are in turn dense in their respective closures which we denote $\mathcal{G}_H(\Omega) \subseteq \mathcal{K}(\Omega)$ and $\mathcal{G}_W(\Omega) \subseteq \mathcal{P}(\Omega)$. Thus by Lemma 2.2 and Theorem 3.3 we have that $\mathcal{M}_{PN}^{\sigma;\tau}$ and $\mathcal{M}_{DS}^{\sigma;\tau}$ are dense in $\mathcal{C}(\mathcal{G}_H(\Omega))$ and $\mathcal{C}(\mathcal{G}_W(\Omega))$. Finally, by Lemma 2.1 we have the desired result. $\qquad\square$

# 4 LIMITATIONS OF POINTNETS AND DEEPSETS

Note that unlike the classical universal approximation theorem we should not expect to be able uniformly approximate $\mathcal{C}(\mathcal{F}_H(\Omega))$ or $\mathcal{C}(\mathcal{F}_W(\Omega))$ since their elements might not even be bounded functions. For example, $\alpha_K(A) = d_H(A, K)^{-1}$ is unbounded but continuous on $\mathcal{F}_H(\Omega)$ whenever $K \in \mathcal{K}(\Omega)$ but $K \notin \mathcal{F}(\Omega)$. Subtler still, we do not even obtain all elements of $\mathcal{C}_b(\mathcal{F}_H(\Omega))$ and

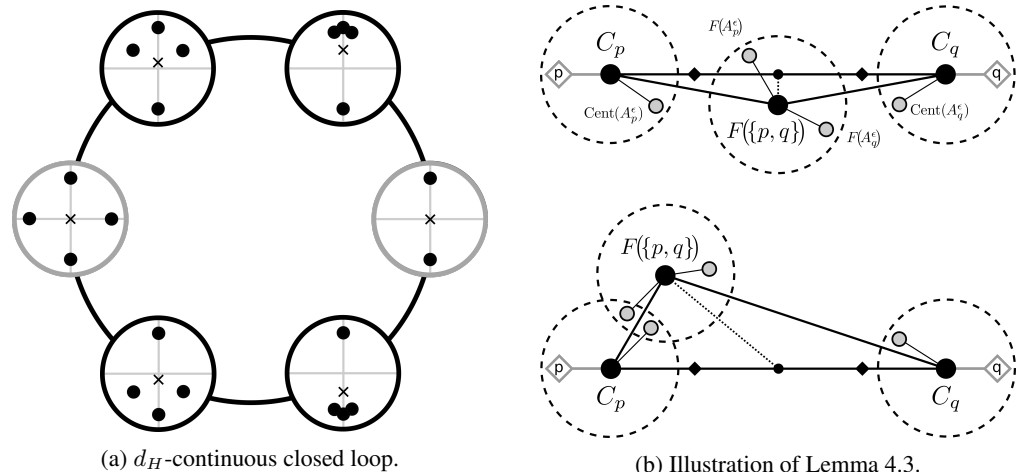

(a) $d_H$-continuous closed loop.

(b) Illustration of Lemma 4.3.

Figure 1: In (a) we see a $d_H$-continuous loop in $\mathcal{F}_{H}^{\leq 4}$ where 'x' marks the center-of-mass. Though the point cloud continuously changes from a 4-point set to a 2-point set, the center of mass discontinuously changes at the moment of convergence. In (b) we see the triangle formed by $C_p, C_q$ and $F(\{p,q\})$ in Lemma 4.3 and how the ball of radius less than $\|C_p - C_q\|/4$ ensures the predicted error for $\|F - \text{Cent}\|$.

$\mathcal{C}_b(\mathcal{F}_W(\Omega))$. As an example of this, observe that $\beta_K = \sin \circ \alpha_K$ is bounded and is also continuous on $\mathcal{F}_H(\Omega)$ because $\alpha_K$ is.

We'll now compare the representation power of these two architectures. Let $\Omega \subseteq \mathbb{R}^N$ be compact. We define the point cloud diameter function $\text{Diam} : \mathcal{F}(\Omega) \to \mathbb{R}$ and point cloud center-of-mass function $\text{Cent} : \mathcal{F}(\Omega) \to \mathbb{R}^N$ by $\text{Diam}(A) = \max_{\boldsymbol{x}, \boldsymbol{y} \in A} d(\boldsymbol{x}, \boldsymbol{y})$ and $\text{Cent}(A) = \frac{1}{|A|} \sum_{\boldsymbol{x} \in A} \boldsymbol{x}$.

**Theorem 4.1.** *Let $(\Omega, d)$ be an infinite compact metric space with no isolated points. Then a function $f : \mathcal{F}(\Omega) \to \mathbb{R}$ is continuous with respect to both $d_H$ and $d_W$ iff it is constant. As a corollary,* $\text{Diam}$ *is uniformly approximable by PointNet networks but not DeepSets networks and* $\text{Cent}$ *is uniformly approximable by DeepSets networks but not PointNet networks.*

*Proof.* Assume $f : \mathcal{F}(\Omega) \to \mathbb{R}$ is both $d_H$-continuous and $d_W$-continuous. Let $A \in \mathcal{F}(\Omega)$ and let $p \in A$. For each $n = 1, 2, \ldots$ choose $A'_n \in \mathcal{F}(\Omega)$ to be an $n$-point set contained within the $1/n$-ball around $p$. We can do this because $\Omega$ is infinite without isolated points. Now let $A_n = A'_n \cup (A \setminus \{p\})$.

Observe that $A_n \overset{d_H}{\to} A$ and $A_n \overset{d_W}{\to} \{p\}$. Thus,

$$f(A) = f\left( \overset{d_H}{\underset{n \to \infty}{\lim}} A_n \right) = \lim_{n \to \infty} f(A_n) = f\left( \overset{d_W}{\underset{n \to \infty}{\lim}} A_n \right) = f(\{p\})$$

Note that $A$ was arbitrary so $f$ must always give a set and any of its singleton subsets the same value. Now let $B, C \in \mathcal{F}(\Omega)$ such that $B \neq C$. Without loss of generality assume $q \in B \setminus C$ and $q \neq r \in C$. Thus by the above,

$$f(B) = f(\{q\}) = f(\{q, r\}) = f(\{r\}) = f(C)$$

thus $f$ must be constant. Conversely, constant maps are always continuous.

Finally, it's known that the $\text{Diam}$ satisfies $|\text{Diam}(A) - \text{Diam}(B)| \leq 2d_H(A, B)$ and hence is $d_H$-continuous on $\mathcal{K}(\Omega)$ and $\text{Cent}$ is $d_W$-continuous on $\mathcal{P}(\Omega)$ because $\text{Ave}_{\pi_i, 0}$ is $d_W$-continuous (here $\pi_i$ is the projection onto the $i$-th component map). This means they are uniformly continuous on $\mathcal{F}_H(\Omega)$ and $\mathcal{F}_W(\Omega)$ respectively and so the result follows from the above and Theorem 3.4. $\qquad \square$

While it is interesting to know that these neural networks describe fundamentally different kinds of functions when we allow for unbounded cloud cardinality, in practice there is always a bound due to computational resource limitations. The next result shows that even when we bound the cloud cardinality, PointNet still cannot uniformly approximate the center-of-mass function.

**Theorem 4.2.** *Let $\Omega \subseteq \mathbb{R}^d$ be an infinite compact set with no isolated points. Then for every $d_H$-continuous $F : \mathcal{F}^{\leq k}(\Omega) \to \mathbb{R}^d$ there exists $A \in \mathcal{F}^k(\Omega)$ such that*

$$\|F(A) - \mathrm{Cent}(A)\| \geq \frac{k-2}{2k} \mathrm{Diam}(\Omega).$$

*In particular, since PointNet architectures are $d_H$-continuous they cannot uniformly approximate center-of-mass when $k \geq 3$ and suffer from the above uniform-norm error lower-bound.*

It is possible to see that PointNet cannot uniformly approximate $\mathrm{Cent}$ by considering Figure 1a. Moving $k-2$ points in a $k$-element cloud to either of the remaining two points produce the same 2-element cloud in a $d_H$-continuous way. By $d_H$-continuity, this means PointNet must output similar centers for the cloud in the top-right and bottom-right – this introduces error. To obtain the explicit error lower bound in Theorem 4.2, we need a slightly more detailed result.

**Lemma 4.3.** *Let $\Omega \subseteq \mathbb{R}^d$ be infinite with no isolated points and $F : \mathcal{F}^{\leq k}(\Omega) \to \mathbb{R}^d$ an arbitrary $d_H$-continuous map. Then for every distinct $p, q \in \Omega$ and $0 < \tau < 1$ there exists an $A \in \mathcal{F}^k(\Omega)$ such that $p, q \in A$ and*

$$\|F(A) - \mathrm{Cent}(A)\| > (1 - \tau)\left(\frac{k-2}{2k}\right)\|p - q\|$$

*In particular, since PointNet architectures are $d_H$-continuous they must satisfy this error bound.*

*Proof.* Assume $k \geq 3$ since the inequality is trivial for $k = 1, 2$. Let $C_p := \frac{p+q}{k} + (\frac{k-2}{k})p$ and $C_q := \frac{p+q}{k} + (\frac{k-2}{k})q$ and observe that $\|C_p - C_q\| = \frac{k-2}{k}\|p - q\|$. Now let $\epsilon = \tau \|C_p - C_q\|/4$ and choose $A_p^\epsilon$ and $A_q^\epsilon$ to be $k$-point supersets of $\{p, q\}$ such that $A_p^\epsilon \setminus \{q\} \subseteq B_\epsilon(p)$ and $A_q^\epsilon \setminus \{p\} \subseteq B_\epsilon(q)$. We can do this because $p$ and $q$ are not isolated points. Since $F$ is $d_H$-continuous and $A_p^\epsilon, A_q^\epsilon \overset{d_H}{\to} \{p, q\}$ as $\epsilon \to 0$, we can additionally demand that $\|F(A_p^\epsilon) - F(\{p, q\})\| < \epsilon$ and $\|F(A_q^\epsilon) - F(\{p, q\})\| < \epsilon$. Next observe that

$$\mathrm{Cent}(A_p^\epsilon) = \frac{p+q}{k} + (1/k)\sum_{a \in A_p^\epsilon \setminus \{p,q\}} a, \qquad \mathrm{Cent}(A_q^\epsilon) = \frac{p+q}{k} + (1/k)\sum_{a \in A_q^\epsilon \setminus \{p,q\}} a.$$

By the triangle inequality it follows that $\|\mathrm{Cent}(A_p^\epsilon) - C_p\| < \epsilon$ and $\|\mathrm{Cent}(A_q^\epsilon) - C_q\| < \epsilon$. Now we can consider the triangle in $\mathbb{R}^d$ formed by $C_p, C_q$, and $F(\{p, q\})$ and realize by basic geometry that one of $\|F(\{p, q\}) - C_p\|$ or $\|F(\{p, q\}) - C_q\|$ must greater than or equal to $\|C_p - C_q\|/2$ (see Figure 1b). Without loss of generality, let $\|F(\{p, q\}) - C_q\| \geq \|C_p - C_q\|/2$. Then,

$$\frac{\|C_p - C_q\|}{2} \leq \underbrace{\left\|F(\{p, q\}) - F(A_q^\epsilon)\right\|}_{<\epsilon} + \left\|F(A_q^\epsilon) - \mathrm{Cent}(A_q^\epsilon)\right\| + \underbrace{\left\|\mathrm{Cent}(A_q^\epsilon) - C_q\right\|}_{<\epsilon}$$

Thus,

$$\left\|F(A_q^\epsilon) - \mathrm{Cent}(A_q^\epsilon)\right\| \geq \frac{\|C_p - C_q\|}{2} - 2\epsilon = (1 - \tau)\left(\frac{k-2}{2k}\right)\|p - q\|.$$

Thus $A_q^\epsilon$ is the promised set which achieves the desired uniform-norm error. □

Theorem 4.2 now follows easily.

*proof of Theorem 4.2.* Choose $p, q \in \Omega$ so $\|p - q\| = \mathrm{Diam}(\Omega)$ and take $\tau \to 0$ in Lemma 4.3. □

## 5 EXPERIMENTS

The proof of Lemma 4.3 not only establishes the error bound but also suggest an algorithmic approach to finding point clouds that exhibit the failure of uniform approximation. This let's us produce adversarial examples to the centor-of-mass problem for PointNet. When $\Omega$ is additionally convex – e.g. the unit disk $D$ in $\mathbb{R}^2$ – it becomes fairly easy to construct many examples of $A_p^\epsilon$ and $A_q^\epsilon$ explicitly for a given PointNet model, allowing us to empirically verify the uniform-norm error lower

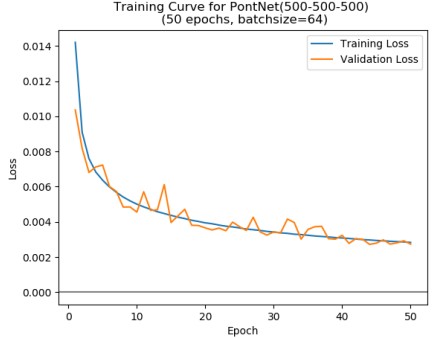

(a) Training curve of a PointNet model learning Cent.

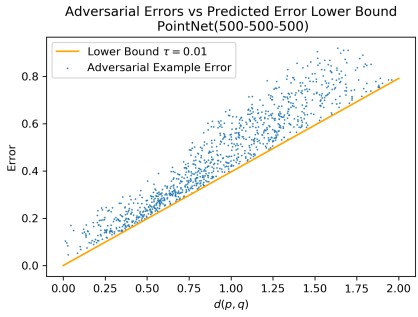

(b) Experimental test of error lower bound.

Figure 2: In (a) we see the training curve for the PointNet model learning the center-of-mass of random 10-element point clouds. In (b) we see that we are always able to find adversarial examples to this task producing errors at least as large as the theoretical gaurantee.

bound. In the following experiment, we train a simple PointNet architecture to learn the center-of-mass for 10-element point clouds in $D$. We train on a synthetic data set of 1 million point clouds (each element uniformly sampled from $D$) labeled with their center-of-mass. The PointNet architecture has 500K trainable parameters. The network has the form $F(A) = \boldsymbol{\rho}(\max_{\boldsymbol{a} \in A} \boldsymbol{\varphi}(a))$ where $\boldsymbol{\varphi}$ has 2-D input layer, 500-D hidden layer, and 500-D linear output layer, and $\boldsymbol{\rho}$ has 500-D input layer, 500-D hidden layer and 2-D linear output layer (in accordance with the Point-Cloud-UAT). The hidden layers of $\boldsymbol{\varphi}$ and $\boldsymbol{\rho}$ are ReLU. Since it is not possible to train with respect to the uniform-norm, we opt for the traditional $L^2$ loss. In Figure 2a we can see the training curve settles after about 50 epochs. The loss at the 50th epoch however is 3-orders-of-magnitude greater than the predict worst case.

To form our adversarial examples, we pick a nonzero $\tau = 0.01$ and two distinct points $p, q \in D$ at random. We set $\epsilon = \tau \|p - q\| / 4$. We then sample $D$ another 8 times. We then linearly pull those 8 points towards $p$ and towards $q$, sufficiently close so that the criteria in the proof of Lemma 4.3 are satisfied. This $\{p, q\}$ adjoined with the 8 points pulled towards $p$ and $q$ form $A_p^\epsilon$ and $A_q^\epsilon$. We are theoretically ensured one of these two will have error larger than our bound. In Figure 2b we plot the the produced adversarial error vs the distance between $p$ and $q$ that were used to make $A_p^\epsilon$ and $A_q^\epsilon$. As predicted, all the adversarial errors lie above the line representing the uniform-norm error lower bound.

## 6    CONCLUSIONS AND FUTURE WORK

The failure of the perceptron model to learn the XOR function was a blow that motivated the search for new models. When classical feed-forward neural networks began to be successful, the spectre of limited representation power loomed over the field until the universal approximation theorem Cybenko (1989)Hornik et al. (1989)Leshno et al. (1993) resolved the question. In this work we resolved the same question for the case of two current deep learning models for point clouds, and laid out a program that could work for other models as well. However, unlike the case of classical neural networks on compact domains of $\mathbb{R}^N$, the same question for point clouds has a less definitive answer even after having determined all of the uniformly approximable functions. Each method explored here has their strengths and limitations but the presence of useful functions out-of-reach of PointNet and DeepSets – even when considering bounded cardinality – opens the door for further research.

There are still many questions. Will merging PointNet and DeepSets in a way to obtain greater approximation power? Are there other useful topologies on $\mathcal{F}(\Omega)$ for which new kinds of continuous neural networks can be constructed? How much do these limitations matter in practice? And finally, are there ways of developing architectures from the ground up with desirable yet different approximation capabilities on point clouds and what are they like?

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

## A APPENDIX

*Proof of Lemma 2.1.* First we show that the $r$ is a linear isometry. Since $X$ is compact for $f \in \mathcal{C}(X)$ there is a $p \in X$ so that $|f(p)| = \|f\|_X$. By density of $D$ there is a sequence $p_n \in D$ that limits to $p$. So

$$\|f\|_X = |f(p)| = \lim_{n \to \infty} |f(p_n)| \leq \sup_{x \in D} |f(x)| = \|r(f)\|_D \leq \sup_{x \in X} |f(x)| = \|f\|_X$$

Next, since $\mathcal{C}(X)$ is complete, so is its isometric image $r(\mathcal{C}(X))$ and because $\mathcal{C}_b(X)$ is complete that means $r(\mathcal{C}(X))$ is closed. Thus,

$$r(\mathcal{C}(X)) = r(\overline{\mathcal{N}'}) \subseteq \overline{r(\mathcal{N}')} \subseteq \overline{r(\mathcal{C}(X))} = r(\mathcal{C}(X))$$

where the first subset results from continuity. Thus $\overline{\mathcal{N}} = \overline{r(\mathcal{N}')} = r(\mathcal{C}(X))$.

Finally, to show $r(\mathcal{C}(X)) = \mathcal{U}(D)$ note that every uniformly continuous function $g$ on $D$ continuously extends to a function on $X$ (because $D$ is dense in $X$) placing this extension in $\mathcal{C}(X)$ and so $g \in r(\mathcal{C}(X))$. The reverse inclusion follows as well because restriction preserves uniform continuity. $\square$

*Proof of Lemma 2.2.* First we show that $\mathrm{Max}_f$ is $d_H$-continuous. Let $\epsilon > 0$. Since $\Omega$ is compact, $f$ is uniformly continuous and so there is a $\delta > 0$ so that $|f(x) - f(y)| < \epsilon/2$ whenever $d(x, y) < 2\delta$. Now let $A, B \in \mathcal{K}(\Omega)$ and suppose $d_H(A, B) < \delta$. By definition this means $A \subseteq B_\delta$ and $B \subseteq A_\delta$. By the triangle inequality we have

$$|\mathrm{Max}_f(A) - \mathrm{Max}_f(B)| \leq |\mathrm{Max}_f(A) - \mathrm{Max}_f(A_\delta)| + |\mathrm{Max}_f(A_\delta) - \mathrm{Max}_f(B)|$$

Since $A_\delta$ is compact there is a $p \in A_\delta$ so that $\mathrm{Max}_f(A_\delta) = f(p)$. Observe that if $q \in K \subseteq A_\delta$ with $d(p, q) < 2\delta$ then $|f(p) - f(q)| < \epsilon/2$ and $f(p) = \mathrm{Max}_f(A_\delta) \geq \mathrm{Max}_f(K) \geq f(q)$. This implies $|\mathrm{Max}_f(A_\delta) - \mathrm{Max}_f(K)| < \epsilon/2$. In particular, since $p \in A_\delta$ there is an $a \in A$ such that $d(p, a) < \delta$, and since $B$ is compact there is a $b \in B$ closest to $p$ and so,

$$d(p, b) = d(p, B) \leq d_H(A_\delta, B) \leq d_H(A_\delta, A) + d_H(A, B) < 2\delta.$$

Thus $|\mathrm{Max}_f(A_\delta) - \mathrm{Max}_f(A)|$ and $|\mathrm{Max}_f(A_\delta) - \mathrm{Max}_f(B)|$ are less than $\epsilon/2$ and so $|\mathrm{Max}_f(A) - \mathrm{Max}_f(B)| < \epsilon$ as desired.

To see why $\mathrm{Ave}_{f,b}$ is $d_W$-continuous recall that the topology of $d_W$ is the same as the weak-* topology for measures and so the map $\mu \mapsto \int f \, d\mu$ is by definition continuous whenever $f \in \mathcal{C}(\Omega)$.

$$\mathrm{Ave}_{f,b}(i_\mathcal{P}(A)) = \mathrm{Ave}_{f,b}\left( \frac{1}{n} \sum_{a \in A} \delta_a \right) = b + \frac{1}{n} \sum_{a \in A} \int f \, d\delta_a = b + \frac{1}{n} \sum_{a \in A} f(a) = \mathrm{ave}_{f,b}(A)$$

It's clear that $\mathrm{Max}_f \circ i_\mathcal{K} = \max_f$. The other identity follows from the linearity of integration.

Lastly, by composition of continuous functions it follows that PointNet and DeepSets continuously extend to $\mathcal{K}(\Omega)$ and $\mathcal{P}(\Omega)$. Since $(\Omega, d)$ compact implies both $\mathcal{K}(\Omega)$ and $\mathcal{P}(\Omega)$ are compact, we can deduce that PointNet and DeepSets are uniformly continuous on $\mathcal{F}_H(\Omega)$ and $\mathcal{F}_W(\Omega)$. $\square$

