# OpenReview forum: "Limitations for Learning from Point Clouds"
_ICLR.cc/2020/Conference — Reject_

### Official Review · AnonReviewer2 · 2019-10-18
**Official Blind Review #2**

**Rating:** 8

**Review:**

PointNet (Qi et al, 2017) and Deep sets (Zaheer et al, 2017) have allowed to use deep architectures that deal with point clouds as inputs, taking into account the invariance in the ordering of points. However, existing results on their approximation abilities are limited to fixed cardinalities. This paper removes the cardinality limitation and gives two kinds of results:

1.	PointNet (resp. Deep sets) can approximate uniformly real-valued functions that are uniformly continuous with respect to the Hausdorff (resp. Wasserstein) metric;
2.	Only constant functions can be uniformly approximated by both (PointNet, Hausdorff metric) and (Deep sets, Wasserstein).

This paper brings a valuable theoretical contribution to the existing state of the art of their approximation abilities. With some improvements, I am willing to increase the score.

1.	The introduction lacks insight into the literature on point cloud or measure networks, including in practice, which would motivate the subject and place it more precisely within the literature.
2.	Notations in the section 2.4 make the reading particularly unclear. Notations should showcase the result that theoretically, only two hidden layers (with appropriate definitions) are needed.
3.	The paper lacks an experimental section. It would be interesting to investigate empirically the limitations of these architectures, for instance by playing with the diameter and center of mass functions as suggested in 3.3.

- Post rebuttal: the authors have addressed 3., therefore I am leaning towards accept.

**Experience Assessment:**

I have published one or two papers in this area.

**Review Assessment: Checking Correctness Of Derivations And Theory:**

I assessed the sensibility of the derivations and theory.

**Review Assessment: Checking Correctness Of Experiments:**

N/A

**Review Assessment: Thoroughness In Paper Reading:**

I read the paper at least twice and used my best judgement in assessing the paper.

---

> ### Author Response · Authors · 2019-11-15
> **Regarding some of these issues**
>
> Thank you for careful review and insights.
>
> Due to time constraint in obtaining newly proven error bound (added in revision) and its experimental verification for in the bounded cardinality case  -- which we hope addresses your concerns in 3) -- we were not able to fully address 1). If there remains time to edit after this comment we would certainly like to expand on this.
>
> Regarding 2), we carefully considered a change of notation but found that it would make the management of the proofs in the paper more unwieldy and possibly more opaque in other regards. The notation is also there to help illustrate how sparse the architecture can be while preserving the UAT and to distinguish between the two types of layers involved with neural networks on these more difficult spaces.
>
> Regarding 3) we made an official comment at the top that we believe addresses these concerns.

---

### Official Review · AnonReviewer1 · 2019-10-23
**Official Blind Review #1**

**Rating:** 3

**Review:**

This work examines the fundamental properties of two popular architectures -- PointNet and DeepSets -- for processing point clouds (and other unordered sets). The authors provide a new universal approximation theorem on real-valued functions that doesn't require the assumption of a fixed cardinality of the input set. They further provide examples of functions that can't be mutually approximated by PointNets and DeepSets.

- It is important in order to know the representational power and fundamental limitations of basic algorithmic building blocks. However, for the two base architectures that are examined in this work, UATs where already provided in the original manuscripts. The presentation in this paper does remove the assumption of a fixed cardinality, but since this seems to be a mild assumption, it is not clear what is gained by this (beyond mathematical elegance). The paper doesn't give any hints here.

- The paper shows two specific functions, where one can be approximated by PointNet but not by DeepSets and vice-versa. The authors again note that this might vanish when a fixed cardinality of the input point cloud is assumed. Again, it seems that this is a mild restriction and I'd like the authors to elaborate on the importance of removing this restriction.

- The paper is clearly targeted at a specialist audience and invokes dense and advanced mathematical concepts. It might find a better audience at a venue that is more specialized in this type of work (e.g. applied mathematics or more theory-focused machine learning venues).

Summary: UATs are important and interesting, however, they do exist for the architectures that are targeted in this paper. It is not clear what is gained by the main difference, i.e. removing the cardinality. I'd be grateful if the authors could comment on this.

Disclaimer: While I believe to have a reasonable mathematical background, I'm not an expert in this field and my assessment is primarily based on the bottom line of these proofs.

=== Post rebuttal update ===
I'd like to thank the authors for their efforts and additional insights. The additional illustration improves the accessibility of the paper. However, the rebuttal does not alleviate my concerns about additional impact beyond the UATs in the original paper. I thus maintain my recommendation of weak reject

**Experience Assessment:**

I do not know much about this area.

**Review Assessment: Checking Correctness Of Derivations And Theory:**

I did not assess the derivations or theory.

**Review Assessment: Checking Correctness Of Experiments:**

N/A

**Review Assessment: Thoroughness In Paper Reading:**

I read the paper at least twice and used my best judgement in assessing the paper.

---

> ### Author Response · Authors · 2019-11-15
> **Response to some of the above concerns**
>
> Thank you for your detailed review. Below we attempt to address some of your points:
>
> "However, for the two base architectures that are examined in this work, UATs where already provided in the original manuscripts."
>
> Though it is true that the original manuscripts provide UATs, it is not a completely apples-to-apples comparison in the case of DeepSets. As mentioned in the paper, we made a slight adjustment to the form of the DeepSets architecture by replacing summation with averaging. We believe this variation is not directly addressed by the theorem presented in the original manuscript. We also believe replacing summation with averaging provides a theoretical improvement by allowing one to interpret DeepSets as acting continuously on probability measures, which lends some intuition to its behavoir when applied to larger point clouds.
>
> Our work also elucidates how simplistic the architecture can be while retaining UATs.
>
> "The paper shows two specific functions, where one can be approximated by PointNet but not by DeepSets and vice-versa. The authors again note that this might vanish when a fixed cardinality of the input point cloud is assumed. Again, it seems that this is a mild restriction and I'd like the authors to elaborate on the importance of removing this restriction."
>
> We added new results that demonstrate this problem does not go away when cardinality is bounded. At least not for the case of PointNet trying to approximate the center-of-mass. We are able to extract lower bounds on the uniform-norm error (independent of the details of the PointNet implementation) and are able to generate as many examples as we desire that achieve error above a theoretical guarantee. This forms the basis of our empirical verification of our results.
>
> "The paper is clearly targeted at a specialist audience and invokes dense and advanced mathematical concepts. It might find a better audience at a venue that is more specialized in this type of work"
>
> We hope that the addition of an explicit error lower bound and experiment confirming it will broaden the appeal.
>
> "UATs are important and interesting, however, they do exist for the architectures that are targeted in this paper. It is not clear what is gained by the main difference, i.e. removing the cardinality. I'd be grateful if the authors could comment on this. "
>
> We believe that in addition removing the cardinality assumption, and elucidating how simple these networks can get away with being, the generalization of the classical UAT to compact Hausdorff spaces could prove useful to other researchers working with other non-traditional data types. In particular, the proofs in the original manuscripts of PoitnNet and DeepSets use techniques that don't seem to readily apply to the other, let alone scale to arbitrary finite cardinality. The introduction of this fairly easy generalization of the classical UAT is what allowed us to prove multiple universal approximation theorems with one method of attack.

---

### Official Review · AnonReviewer3 · 2019-10-27
**Official Blind Review #3**

**Rating:** 3

**Review:**

The paper aims to establish novel theoretical properties of known point-based architectures for deep learning, PointNet and DeepSets. To this end, the authors prove a series of theoretical results and establish limitations of these architectures for learning from point clouds.

Unfortunately, due to a delay with my review, I cannot afford the joy of carefully examining the proofs.

I do not find significant practical value in the obtained results. While the theorems proved in the paper are original and novel, they are a refinement of the already known results regarding approximation theorems for PointNet and DeepSets, respectively, hence only a marginal improvement in understanding these function classes.

**Experience Assessment:**

I do not know much about this area.

**Review Assessment: Checking Correctness Of Derivations And Theory:**

I did not assess the derivations or theory.

**Review Assessment: Checking Correctness Of Experiments:**

I did not assess the experiments.

**Review Assessment: Thoroughness In Paper Reading:**

I made a quick assessment of this paper.

---

> ### Author Response · Authors · 2019-11-15
> **Improvements to results allow for the generation of adversarial examples to PointNet when learning center-of-mass**
>
> Thank you for your review.
>
> Before we were unable to provide experiments due to the difficulty of algorithimically finding examples that achieve L^\infty errors promised by the inability to uniformly approximate. This was made even more difficult because point cloud cardinality was free to be unbounded. However we now have results that apply to the bounded cardinality case for PointNet and show that this still cannot uniformly approximate the center-of-mass. We provide an explicit error bound and empirical test of this bound in the paper. We found these results to be robust in the sense that we could randomly sample pairs of points, and use those as a seed to find sets that yield the predicted worst case error. In a sense, this demonstrates a method to produce adversarial point cloud examples with theoretical guarantees.
>
> We hope this further elucidates the nature of these function classes and improves the practicality of these results.

---

### Author Response · Authors · 2019-11-15
**Addition of Theorem & Explicit Error Bound in Case of Bounded Cardinality and Empirical Verification**

We thank the reviewers for their thoughtful and constructive comments. We address this comment to the three reviewers who shared similar concerns regarding the practicality of the results for unbounded cardinality and lack of experiments. We took these concerns seriously and pushed the methods to arrive at no-go theorems for the case of bounded point cloud size. We prove that even with cloud size bounded to k or less points, PointNet cannot uniformly approximate the center-of-mass function. Moreover we prove an explicit worst case error lower bound (worst case error is at least (k-2)*Diam(domain)/k)  and empirically confirm the robustness of these results. We also add images to aid in the understanding of these results.

We hope that these changes will alleviate some of the concerns of the reviewers. We will individually address the remaining concerns in the comments.

---

### Decision · Program_Chairs · 2019-12-19

**Decision:**

Reject

**Comment:**

The present paper establishes uniform approximation theorems (UATs) for PointNet and DeepSets that do not fix the cardinality of the input set.

Two nonexperts read the paper and came away not understanding what this exercise has taught us and why the weakening of the hypotheses was important. The authors made no attempt to argue these points in their rebuttals and so I went looking at the paper to find the answer in their revisions, but did not find it after scanning through the paper. I think a paper like this needs to explain what is gained and what obstructions earlier approaches met, and why the current techniques side step those. One of the reviewers felt that the fixed cardinality assumption was mild. I'm really not sure why the authors didn't attack this idea. Maybe it is mild in some technical sense?

What I read of the paper seemed excellent in term of style and clarity. I think the paper simply needs to make a better case that it is not merely an exercise in topology. I think the result here is publishable on its own grounds, but for the paper to effectively communicate those findings, the authors should have revised it to address these issues. They chose not to and so I recommend ICLR take a pass. Once the reviewers revised the framing and scope/impact, provided it doesn't sound trivial, I think it'll be ready for publication.